# Fostering Empathy Through Play: The Impact of *Far From Home* on University Staff’s Understanding of International Students

**DOI:** 10.3390/bs15060820

**Published:** 2025-06-14

**Authors:** Shuanghui Sofia Shan, Sam Illingworth

**Affiliations:** 1School of Education, Nanchang Vocational University, Anyi, Nanchang 330500, China; sofiashantabares@outlook.com; 2Department of Learning and Teaching Enhancement, Edinburgh Napier University, Edinburgh EH11 4QQ, UK

**Keywords:** games, empathy, international students, student experience, wellbeing, belonging

## Abstract

This study investigates the potential of *Far From Home*, a non-digital board game, as an innovative tool for fostering empathy among university staff towards international students. International students face multifaceted challenges—linguistic barriers, cultural dissonance, and systemic inequities—yet traditional staff training often fails to cultivate the perspective-taking required for meaningful support. Using a mixed-methods approach, we analysed data from 82 participants across 10 game sessions, including surveys (n = 27), recorded gameplay observations, and semi-structured interviews (n = 6). Thematic analysis explored how role-playing as student avatars and collaborative problem-solving influenced staff empathy. The results demonstrated the game’s effectiveness in bridging cultural gaps, with participants reporting a heightened awareness of structural barriers and reduced stereotyping. Notably, the emergent findings suggested a “contrast commitment” effect, where witnessing biassed behaviours reinforced staff’s dedication to equitable practices. This study advocates for game-based training as a complement to existing programmes, with future research needed to assess longitudinal impacts. Potential applications include adapting the framework for other marginalised student groups and institutional contexts.

## 1. Introduction

International students embarking on higher education navigate a complex and often demanding landscape, fraught with intersecting challenges that extend beyond the purely academic. Linguistic marginalisation in academic writing, as highlighted by [42] ([42]), presents a significant hurdle, where the subtle nuances of academic discourse, disciplinary-specific vocabulary, and expectations around argumentation can create substantial barriers for non-native English speakers. This can manifest in difficulties with understanding assignment instructions, articulating complex ideas in written form, and meeting the stylistic conventions of academic essays and theses, leading to feelings of inadequacy and impacting academic performance ([36]; [37]).

Furthermore, cultural dissonance in pedagogical expectations, explored by [30] ([30]), adds another layer of complexity. Differing educational norms regarding student–teacher relationships, classroom participation, approaches to reading and learning (e.g., independent study versus collaborative work), and assessment methods can lead to misunderstandings, anxiety, and a sense of disorientation for international students accustomed to different educational systems ([18]; [20]; [28]; [46]). For instance, a student from a culture emphasising deference to instructors might be hesitant to engage in critical debate, a common pedagogical practice in many Western universities, potentially leading to lower participation grades and a feeling of being “out-of-sync” with their peers ([26]).

Beyond these academic and cultural adjustments, international students also encounter systemic barriers embedded within institutional policies and practices. [9] ([9]) and [35] ([35]) underscore how seemingly neutral policies can inadvertently disadvantage international students, ranging from complex visa regulations and financial aid limitations to a lack of culturally sensitive support services and an inadequate understanding of their unique needs by administrative staff. Examples of these barriers can include cumbersome processes for extending visas, insufficient financial assistance tailored to international student status, a lack of mental health support that acknowledges cultural differences in expressing distress, and administrative hurdles in accessing essential services due to unfamiliarity with institutional procedures. These systemic challenges can contribute to feelings of isolation, stress, and marginalisation, significantly impacting international students’ overall wellbeing and academic success.

To address these struggles, the non-digital board game *Far From Home* was developed. This game immerses staff in scenarios experienced by international students—such as visa rejections or juggling part-time work with deadlines—to foster a deeper understanding of their daily challenges. This paper elaborates the game’s design and presents the findings from a small-scale evaluation of its use in staff development at a Scottish university. This paper explores how such games can complement existing training methods to foster deeper understanding, critical engagement, and more inclusive institutional practices.

## 2. Literature Review

### 2.1. Traditional and Dialogic Training Approaches

In recognising the struggles faced by international students, there has been a growing emphasis within the higher education sector on providing universities with the skills and understanding necessary to provide effective support ([27]; [33]; [38]). Traditional staff training programmes have evolved over time, with a shift from predominantly didactic methods to more engaging dialogic approaches. While the former can impart knowledge about international student challenges, such as linguistic marginalisation or cultural dissonance, they often lack the active processing and perspective-taking required for genuine understanding ([24]).

Dialogic methods, involving discussions, Q&A sessions, and shared reflections, represent a step forward in engaging trainees cognitively ([45]). By encouraging interaction and the exchange of viewpoints, dialogic workshops can facilitate a greater understanding of diverse perspectives compared to purely didactic approaches. However, as [2] ([2]) cautions, such methods risk reinforcing a superficial “tourist” approach to multicultural education if they prioritise abstract dialogue over critical engagement with systemic inequities. While dialogic methods encourage cognitive engagement, they may not fully replicate the lived experience or generate the same level of emotional and cognitive empathy as immersive, experiential learning approaches like game-based learning, which allow learners to directly experience another’s perspective.

### 2.2. Game-Based Learning for Empathy Development Through Simulation

Game-based learning (GBL) offers a dynamic alternative to traditional training by simulating real-world experiences. A growing body of research underscores games’ efficacy as empathy-building tools. Non-digital games like *EmPATHs* ([32]) enhance empathy and raise the awareness of the challenges faced by vulnerable groups, engaging diverse audiences from students to practitioners. Similarly, simulation-based methodologies—including role-play and games—prove effective in teaching empathy. [4]’s ([4]) systematic review of 27 studies found simulation the most beneficial when learners “stand in patients’ shoes,” with role reversal (e.g., rotating patient–provider roles) emerging as a critical mechanism for developing empathy.

This aligns with digital games like *The Walking Dead* ([41]), which leverages its zombie-apocalypse narrative to immerse players in ethically charged dilemmas. Despite its fictional setting, its choice-driven mechanics force engagement with human values, fostering deep cognitive and emotional investment ([21]). [39] ([39]) demonstrated its pedagogical value by using the game’s “magic circle” ([34]; [40])—a psychologically safe space—to facilitate moral education. Students made difficult decisions in low-stakes scenarios, enabling ethical reasoning without real-world repercussions ([5]).

*Far From Home* builds on these principles, using structured role-play scenarios (e.g., visa rejections, academic exclusion) to simulate international students’ lived experiences. Like *EmPATHs* and *The Walking Dead*, it leverages the “magic circle” for low-risk experimentation, while [4]’s ([4]) findings on role reversal directly inform its design of staff-as-student perspective-taking.

### 2.3. Theoretical Foundations of Empathy

GBL’s pedagogical approach rests on a nuanced understanding of empathy itself. Empathy is not a single construct but a multidimensional interplay of cognitive empathy—the ability to adopt another’s perspective—and emotional empathy—the capacity to share in their feelings ([12]). Importantly, empathy differs from compassion, which involves concern without necessarily understanding the other’s standpoint ([3]). In educational contexts, cognitive empathy is particularly valuable, shifting staff responses from well-meaning sympathy to informed, structural understanding. For example, staff who interpret silence as disengagement may fail to recognise it as a culturally respectful behaviour ([18]) stemming from differing educational norms regarding classroom participation, as highlighted by [30] ([30]). Those attuned to cognitive empathy can better scaffold inclusive responses, bridging this cultural dissonance.

Yet emotional empathy has its limits. Over-identification—“I feel your pain”—can inadvertently centre the staff member’s perspective and obscure the structural nature of the problem ([11]). Sustainable empathy in higher education thus requires a careful balance: connecting emotionally while maintaining a critical lens on institutional norms and practices.

Empathy is not merely a soft skill. It has direct implications for addressing systemic barriers and enhancing inclusion, student wellbeing, and academic success. Empathetic staff interactions foster a sense of belonging, which is closely tied to persistence and achievement ([16]). Dismissive responses to linguistic challenges—for instance, labelling non-native writing as “unscholarly”—can reinforce institutional alienation ([42]; [36]). Conversely, staff who adopt a cognitive empathetic stance can reframe perceived deficits as unfamiliarity with Eurocentric academic norms, allowing staff to critique those norms and scaffold more inclusive forms of success ([25]).

This review highlights the evolution of staff training towards more engaging methods, the unique potential of game-based learning for fostering empathy, and the multidimensional nature of empathy itself. Despite advancements, there remains a need for targeted, immersive interventions that move beyond abstract understanding to truly cultivate cognitive empathy and address the systemic barriers faced by international students. The subsequent sections detail how *Far From Home* was designed and evaluated to meet this critical need, building on the theoretical foundations and pedagogical principles discussed herein.

## 3. Game Design

The non-digital board game *Far From Home* was developed as an immersive training intervention to foster a deeper understanding of and empathy for the multifaceted experiences of international students among university staff. Its design is rooted in the principles of game-based learning, experiential education, and empathy development, operationalising these through three core mechanics.

### 3.1. Projective Identity and Intersectional Avatars

The game encourages players to engage in projective identity ([17]) by creating avatars with intersectional identities. Informed by [19] ([19]) figured identities framework, the game’s nationality cards and associated prompts are designed to guide players in building complex identities (incorporating age, gender, class, family obligations, and academic focus) rather than relying on reductive cultural assumptions. This mechanic aims to require players to reconcile their own preconceptions with the lived complexity of their character’s background and experiences, fostering a deeper, empathetic connection from the outset.

### 3.2. Experiential Learning Through Scenario Mapping

The game’s scenarios are mapped onto the structure of a typical master’s programme, embedding [23]’s ([23]) experiential learning cycle directly within the gameplay. This involves a design where players are intended to encounter concrete experiences (e.g., simulated challenges like group work exclusion due to accented English or ambiguous assignment instructions rooted in research on real student challenges; [42]). These experiences are designed to be followed by structured reflective prompts, leading to a conceptual critique of institutional practices that may inadvertently disadvantage international students (e.g., issues of digital exclusion). Finally, players are intended to engage in active experimentation by collectively devising alternative strategies and institutional responses. This cyclical approach aims to anchor abstract empathy in realistic academic struggles, promoting continuous learning and adaptation.

### 3.3. Creating a Safe Space for Dialogue and Structural Critique

Quest cards leverage the “magic circle” of play ([34]; [40]) to create a psychologically safe space for low-stakes dialogue among diverse staff participants—including academic, administrative, and professional services staff. Within this safe but resonant context, participants are intended to rehearse responses to structural inequities ([43]). Some quests are designed to explicitly foreground systemic frustrations (e.g., system lockout due to delayed tuition payments, referencing systemic barriers highlighted by [35]), while others invite the sharing of celebratory moments and successful coping strategies (e.g., successful use of peer proofreading). These exchanges are designed to help shift the narrative from one of isolated student resilience to one of shared institutional responsibility ([2]), facilitating the flattening of hierarchies among staff and promoting collaborative problem-solving.

Together, these design elements foster a form of empathy that is both emotionally attuned and critically informed. Rather than promoting awareness alone, *Far From Home* invites participants to feel, think, and act from within the student experience. In doing so, it serves as a pedagogical bridge between empathy and institutional change—a necessary shift if higher education is to meet the needs of its increasingly international communities.

## 4. Materials and Methods

This section outlines the methodology employed to assess the impact of the board game *Far From Home* on fostering empathy among university staff towards international students. A mixed-methods approach was adopted to generate a layered understanding of the game’s influence, combining quantitative and qualitative methods. Ethical approval was granted by the university’s research ethics committee prior to the commencement of this study.

The game *Far From Home* was piloted in two playtesting rounds with volunteer staff from a range of departments. These initial sessions were used to refine the game’s mechanics, test the clarity of instructions, and optimise the overall user experience. Revisions were made based on the feedback received, and the final version of the game was launched in March 2024. The development was informed by prior institutional research into awarding gaps, which highlighted the specific structural and pedagogical barriers faced by international students. The game was designed by the lead author in their capacity as an institutional researcher in the Department of Learning and Teaching Enhancement, with the intention of creating a tool to encourage empathetic and critical engagement with these issues.

### 4.1. Participants

A total of 82 university staff members participated across ten scheduled game sessions conducted between March and July 2024. Participation was voluntary and drawn from across academic, administrative, and support roles, representing a broad range of university roles.

Staff who signed up for the game session were organised into groups of 4–6 players per table, with up to four groups per session. Each group consisted of 3–4 avatars representing international student personas. In some cases, two players collaborated to role-play as a single avatar—a design choice that actively encouraged dialogue and collective problem-solving during gameplay (see Figure 1).

### 4.2. Typical Session’s Organisation and Gameplay

A typical *Far From Home* session lasts about 1.5 h, and it involves a structured sequence of activities to ensure a consistent and immersive experience for participants. Each session began with an introduction to the game’s overall objectives and rules. This was followed by the avatar setup phase, a crucial initial step where players would select a nationality card and create a unique international student avatar (see Figure 2). Players were encouraged to enrich their avatar’s identity by assigning them a name, gender, social class, family background, and academic details such as their master’s programme of study and personal purpose for pursuing it. This phase was designed to foster initial personal investment and the adoption of a specific student perspective.

Following avatar creation, players navigated the game board (see Figure 3), which visually resembled a typical master’s study experience. Avatars, represented by meeples, took turns to roll dice and progress towards graduation, the game’s objective. When an avatar landed on a space marked “Q”, the player(s) drew a quest card (see Figure 4a–c) from the stack. These cards presented players with specific, realistic scenarios reflecting international student challenges or opportunities, prompting discussion and decision-making for their avatars. The game provided detailed instructions printed as a handout for the players to follow, and a facilitator (i.e., the Principal Investigator of the project) was present to clarify rules, make judgements, guide discussions, and prompt deeper reflection on the gameplay’s implications for real-world university practices throughout the session.

### 4.3. Data Collection

To evaluate the impact of the intervention, three modes of data collection were used. First, participants were invited to complete an anonymous online survey at the end of each session, accessed via a QR code printed on the game instructions. The survey, developed in Microsoft Forms, included both Likert-scale questions and open-ended prompts to assess participants’ reflections on the design and facilitation of the session, as well as their engagement with the themes raised by the game. The questionnaire was purpose-built for this study, drawing on principles from reflective learning and empathy research (e.g., [23]; [24]), but it was not adapted from a previously validated instrument. Of the 82 participants, 27 submitted responses.

Second, one of the game sessions was audio-recorded and later transcribed. While not a focus group in the traditional sense, this recorded session served as a performative observation, capturing how participants took on character roles and interacted with the game scenarios. The data offered illustrative insights into how staff enacted empathy in situations, adopting character identities and responding to challenges from the perspective of international students. Examples include in-character expressions of struggle and disorientation, such as a player introducing themselves as “Evan, from Nigeria,” and describing fictional difficulties with the weather or interpersonal tensions affecting sleep—articulations that both mirrored and refracted real experiences.

Third, all participants who completed the survey were given the option to volunteer for a follow-up interview by submitting their email address along with the survey. Six participants took part in semi-structured interviews. These provided more in-depth perspectives on their engagement with the game and any perceived shifts in their understanding of the international student experience.

### 4.4. Data Analysis

Quantitative survey data were analysed using descriptive statistics, specifically the mean and standard deviation, to identify trends in participant feedback. Qualitative data from the open survey responses, interview transcripts, and the observational session were analysed thematically following [6]’s ([6]) approach. This process involved familiarisation with the data, coding, theme development, and iterative refinement to identify patterns in how staff engaged with the game and reflected on their roles within the institution.

### 4.5. Researcher Positionality and Ethics

The researcher’s dual position as game designer and evaluator shaped both the development and analysis processes. This positionality offered deep insight into the game’s theoretical underpinnings and design logic but also introduced interpretive proximity. Rather than claiming objectivity, this study employed a reflexive approach, acknowledging the influence of the researcher’s prior knowledge and interpretive lens. This included ongoing critical reflection on assumptions and potential bias throughout the analysis. The resulting interpretation presents a situated understanding of the game’s potential to foster empathetic engagement and critical institutional reflection.

All participants received detailed information about the aims of this study, the voluntary nature of their involvement, their right to withdraw at any time without penalty, and the steps taken to protect their anonymity. Informed consent was obtained in writing and verbally for participation in the survey, individual interviews, and the recorded session. Data were stored securely in a password-protected university-managed drive. Audio files were transcribed using Microsoft Word’s inbuilt transcription tool and deleted following verification. To ensure confidentiality, pseudonyms were used throughout the analysis and reporting. This ethical process applied to all user participants and was introduced prior to data collection, aligning with the broader methodological approach described earlier in this section.

## 5. Results

This section presents the findings from the evaluation of the board game *Far From Home*, focusing on its perceived impact in terms of cultivating empathy among university staff towards international students. The results are presented in two main parts: first, a summary of quantitative survey data providing an overall picture of participant perceptions and game experience and second, a detailed qualitative analysis. The qualitative findings are organised into four overarching themes, derived from a thematic analysis of semi-structured interviews, focus group transcripts, and open-ended survey responses. These themes illustrate the perceived impact of the game in cultivating empathy, facilitating perspective exchange, creating safe spaces for exploration, and revealing institutional bias concerning international students. Direct quotations from participants are used to provide contextualised evidence for the interpretations presented and are situated within the theoretical framework outlined earlier.

### 5.1. Overall Participant Perceptions: Quantitative Survey Findings

The game was overwhelmingly well-received by 27 participants who completed the post-session survey, with the results indicating a highly positive and insightful experience (see Table 1). A significant majority of participants (92%) rated their overall gaming experience as 4 or 5 out of 5, clearly demonstrating its engaging and enjoyable nature. This positive reception was further supported by high mean scores for statements like “I find the game session well organised” (M = 4.63; SD = 0.49) and “I think the game is fun” (M = 4.67; SD = 0.55), reinforcing its perceived quality and enjoyability.

Crucially, the game proved highly effective in achieving its core objective of fostering empathy and understanding. Participants overwhelmingly agreed that “Playing the game makes me understand more of international students’ struggles studying and living in the UK” (M = 4.41; SD = 0.68), with only one participant expressing disagreement. This strong consensus suggests that the game successfully immersed players in the realities faced by international students. Additionally, participants generally found the game’s instructions clear (M = 4.15; SD = 0.70) and did not perceive it as too difficult (M = 1.81; SD = 0.66), contributing to a smooth and accessible learning experience. Interestingly, the statement “Playing the game reminds me of my time studying at school or university” received a lower mean score (M = 2.59; SD = 1.14), perhaps indicating that the simulated international student experience successfully differentiated itself from the participants’ own prior educational journeys.

### 5.2. Cultivating Empathy Through Experiential Engagement

This theme explores how the game’s experiential design helped foster empathy, particularly by enabling staff to engage with the multifaceted challenges international students face, aligning with [23]’s ([23]) experiential learning cycle.

#### 5.2.1. Unveiling the Complexities of Pre-Arrival Experiences

Participants reported a newfound awareness of the bureaucratic and financial hurdles students face prior to arrival, often previously overlooked or misunderstood. One professional services staff member in the interview reflected the following:
“I didn’t know about all the different reporting procedures, like with immigration… students had to have a certain amount of money in their bank account before they could even come. [The game] gave me an appreciation of how dedicated these students are”.

This shift in perception, supported by [9] ([9]) and [35] ([35]), illustrates how procedural empathy ([22]) was engaged through in-game mechanics such as resource management and event cards. These mechanisms simulated the emotional and logistical burden of preparing to study abroad, prompting the recognition of the structural inequities faced by international students. By mirroring real-world bureaucratic hurdles, the game operationalised Kolb’s *concrete experience* phase, grounding abstract challenges in tangible, emotionally resonant scenarios.

#### 5.2.2. Recognising Hidden Struggles and Intersectional Realities

The avatar system, based on [19] ([19]) figured identities framework, encouraged participants to engage with intersectional experiences. One participant in the interview reflected the following:
“If I find colleagues off work with sickness absence, you just assume they’re sick. But actually, there’s other things going on—personal issues, caring responsibilities”.

This insight demonstrates a broadening of awareness from simplistic assumptions to a more layered understanding of individual contexts as members of multiple communities, which echoes [7]’s ([7]) intersectional pedagogy. The game’s design compelled participants to navigate overlapping identities (e.g., international student, caregiver), fostering empathy through the *reflective observation* of systemic barriers beyond academic life.

#### 5.2.3. The Power of Shared Narratives in Building Empathy

Participants valued the chance to share and hear peer experiences during gameplay, reinforcing the social dimension of empathy development. One survey participant noted the following:
“Hearing stories from others… based on their contacts with students… adds more perspectives. It’s a good way of learning from experiences others have had”.

These collective narratives supported vicarious learning and intercultural competence through dialogue ([10]), with the game’s magic circle ([40]) enabling the safe exchange of personal and professional insights.

### 5.3. Facilitating Perspective Exchange Through Role-Play

This theme examines the role of embodied simulation and collaborative play in prompting perspective shifts.

#### 5.3.1. Embodying Student Identities to Understand Their World

Participants reported that taking on student avatars deepened their emotional and cognitive understanding. One interview participant commented the following:
“Playing the role of students helps you think more about their experiences, feelings, and perspectives. You have to imagine how they might feel when explaining situations to others”.

This engagement aligns with [17]’s ([17]) projective identity and [15]’s ([15]) critical play, where inhabiting another’s role foregrounds systemic inequalities. The game’s mechanics—such as role-specific constraints and decision-making dilemmas—bridged the gap between abstract theory and lived reality, enabling participants to “feel” inequities rather than merely recognise them.

#### 5.3.2. Fostering Collaborative Dialogue and Challenging Assumptions

The co-operative aspect of the game encouraged dialogue and challenged assumptions through peer interaction. One survey participant stated the following:
“The game encouraged discussion and challenged views. Peer learning from colleagues was very helpful”.

This finding reflects [43]’s ([43]) concept of communitas and [44]’s ([44]) emphasis on social learning, highlighting the game’s value as a platform for collective sense-making. By structuring interactions around shared goals, the game transformed individual reflections into communal critiques of institutional norms, reinforcing the interdependence of perspective-taking and equity work.

### 5.4. Creating Safe Spaces for Low-Stakes Exploration

This theme describes how the game provided a psychologically safe environment for testing ideas and reflecting on professional roles.

#### 5.4.1. Enabling Risk-Free Problem-Solving and Innovative Thinking

Participants noted the benefit of experimenting without consequences. One academic staff in the interview remarked the following:
“The decisions you make in the game… have no real risk. You can take risks or think differently compared to day-to-day work”.

This underscores the importance of ludic environments for creative engagement ([1]) and the psychological safety of the magic circle ([34]; [40]). The game’s abstraction from real-world repercussions allowed participants to interrogate biases and rehearse equitable practices, fostering a mindset of iterative learning over punitive perfectionism.

#### 5.4.2. Facilitating Reflective Practice Through Dual Role-Playing

The game also prompted reflection on participants’ professional practices. One interview participant described the following:
“By offering advice to students, you think about your own experiences as an academic. It brings both perspectives together”.

This quote demonstrates a form of dialogic reflection ([45]) and aligns with [29]’s ([29]) transformative learning, where perspective-taking leads to potential behavioural change. The dual role-playing structure—alternating between student avatars and advisors in real life—created a feedback loop, enabling participants to critique institutional norms while re-evaluating their own complicity within them.

### 5.5. Institutional Mirroring—Revealing Biases and Motivating Reflexive Change

The board game functions as a pedagogical mirror, reflecting institutional biases while creating spaces for staff to reconcile discrepancies between their egalitarian ideals and colleagues’ stereotypical practices.

#### 5.5.1. Exposing the Need for Empathy-Driven Training

An interview participant, Korsikoff (pseudonym), recounted their discomfort with colleagues’ attitudes during the game:
“There was quite a lot of stereotyping of students… discussing Nigerian students, they implied all behave a certain way. It was verging on racist”.

By homogenising Nigerian students into a monolith, staff participants reproduced structural hierarchies that position international students as perpetual outsiders ([31]). This aligns with [8]’s ([8]) critique of “imposed identities” (p. 17), where individuals are ascribed fixed traits (e.g., nationality, ethnicity) that erase intersectional complexities. 

The participant’s shock at colleagues’ remarks reflects a broader systemic issue: staff often lack an awareness of how unconscious biases shape their support practices. Later, the participant elaborated on this disconnect:
“I was struck by the difference of opinion in different departments… if we had that approach with our students, they would complain about us all the time… I’m glad we don’t treat our students like that… it reemphasised to me the need for this training for staff”.

Here, the game acts as a revelatory intervention, exposing inconsistencies in student support across departments. By simulating real-world scenarios (e.g., stereotyping Nigerian students), it compels staff to recognise the urgency of replacing cultural scripts with empathy-driven practices.

#### 5.5.2. Contrast Commitment—A Reflexive Mechanism for Sustaining Equity

This study introduces the concept of “contrast commitment”—defined as the reflexive reaffirmation of equitable practices that occurs when individuals witness incongruities between their own consciously regulated behaviours and others’ unconscious participation in systemic biases. Grounded in [13]’s ([13]) dual-process model (automatic vs. controlled bias) and [14]’s ([14]) habit-breaking framework, the term emerged from participants’ accounts of dissonance and resolution.

Participant Korsikoff provided a rich account of this self-reflection which exemplifies this process:
“I’m glad that we don’t treat our students like that…I was a bit shocking really, that we were in the same institution and there was this staff’s approach to student support… It reemphasised to me the need for this training for staff”.

Witnessing colleagues’ homogenising assumptions (automatic bias) during role-play created cognitive dissonance, which participants resolved by consolidating their commitment to student-centred approaches (controlled regulation). Unlike [14]’s ([14]) focus on individual habit-breaking, “contrast commitment” positions institutional discrepancies as pedagogical tools. The game’s design—which mirrors departmental contrasts in attitudes—transforms dissonance into a catalyst for accountability, advancing equity work by framing such discrepancies as generative rather than obstructive.

## 6. Discussion

The findings of this study suggest that *Far From Home* offers a distinctive approach to fostering empathy amongst university staff towards international students. Quantitative survey responses support this interpretation, with participants rating the game highly in terms of overall experience and its effectiveness in enhancing their understanding of international student challenges. These self-reported gains align with the qualitative accounts presented in this study, reinforcing the game’s potential as a structured empathy-building intervention. While board games have previously been used in empathy-focused education, the novelty here lies in the game’s specific design—its use of intersectional avatars, narrative scenarios focused on pre-arrival and cultural challenges, and structured reflection prompts tailored to the UK higher education context. This experiential format, allowing staff to navigate simulated challenges and embody student identities, supported a deeper understanding of the often-invisible complexities of international student experiences. Role-playing proved effective in encouraging perspective-taking, while the low-stakes environment fostered open dialogue and challenged assumptions. These observations align with this study’s theoretical framing, drawing on experiential learning ([23]), identity transformation ([17]; [19]), and ludic boundary work ([34]; [40]), offering a useful lens for interpreting the game’s impact.

Critically, the fourth theme—Institutional Mirroring: Revealing Biases and Motivating Reflexive Change—highlighted the game’s capacity to expose ingrained stereotyping and structural disconnects in staff attitudes. One particularly rich account from a participant suggested a potential mechanism we termed “contrast commitment”. This observation describes how witnessing discrepancies between colleagues’ biassed behaviours and one’s own egalitarian practices during the game appeared to reflexively reinforce that participant’s commitment to equitable approaches. Grounded in [13]’s ([13]) dual-process model and [14]’s ([14]) habit-breaking framework, this tentative concept suggests a way in which the surfacing of bias within a shared experience can act as a catalyst for strengthening individual resolve towards inclusive practices.

However, it is important to acknowledge that the concept of “contrast commitment” as defined in this study emerged primarily from the detailed reflections of a single participant. While this account provides a valuable and potentially significant insight into the dynamics of bias awareness and commitment to equity, further research with a larger and more diverse sample is necessary to determine the prevalence and generalisability of this phenomenon across different individuals and institutional contexts. Future studies could specifically explore the conditions under which such “contrast commitment” is more likely to occur and its long-term impact on behaviour change.

This study’s broader findings, supported by feedback from 27 participants across surveys, interviews, and observations, underscore the value of game-based learning for empathy development, perspective-taking, and the initial surfacing of institutional biases.

## 7. Conclusions

This study demonstrates that the board game *Far From Home* holds significant potential as a novel and engaging tool for fostering empathy and raising awareness among university staff regarding the multifaceted experiences of international students. The experiential and role-playing elements of the game effectively facilitated perspective-taking, encouraging a deeper understanding of the structural and personal barriers encountered by this student population. Furthermore, the safe and collaborative nature of the gameplay not only created opportunities for institutional critique and reflective practice but also served as a mirror, reflecting existing biases within the institution.

A striking insight from one participant introduced what we tentatively describe as a process of “contrast commitment”—a reflexive response in which encountering biassed behaviour from others during gameplay appeared to strengthen the participant’s own commitment to equitable practice. This moment of dissonance between self and other seemed to reinforce their values rather than undermine them. Although conceptually promising, this finding is grounded in a single account and should be treated as preliminary, warranting further exploration with a broader and more diverse participant base.

Several limitations of this study and the intervention itself warrant consideration. Firstly, participation in the game sessions was voluntary, which likely resulted in a self-selecting sample of staff members who already possessed a degree of awareness or interest in international student support. This raises questions about how to effectively engage a broader staff audience, including those who may not initially recognise the importance of this issue or feel equipped to provide support. Future strategies for disseminating the game and encouraging participation from a wider range of staff are crucial for maximising its potential impact.

Secondly, the development and implementation of *Far From Home* present practical limitations. The creation of such a game requires significant financial support and a substantial time investment in areas such as art and game design, iterative revisions based on playtesting feedback, and the ongoing facilitation of game sessions by the designer to ensure meaningful engagement. Furthermore, the capacity of each game session is limited to approximately 20 players to facilitate effective conversation and ensure that all voices can be heard. Scaling up the intervention to reach a larger proportion of university staff would necessitate considerable resources and logistical planning.

Thirdly, the effectiveness and productivity of each game session were observed to vary depending on the individual players at each table. Factors such as their flexibility and willingness to engage, their comfort levels in sharing personal thoughts and experiences, and their inclination towards deep conversation significantly influenced the outcomes of the gameplay. This inherent variability in group dynamics poses a challenge for ensuring a consistently impactful gaming experience across all sessions. Strategies for enhancing the productivity of the gaming experience, such as more structured facilitation techniques or pre-game activities designed to foster open communication and explicitly address potential biases, could be explored in future iterations to further leverage the mechanism of “contrast commitment”.

Future research could build on these findings by exploring the longitudinal impact of the game on staff attitudes and behaviours, particularly in terms of sustained empathy, changes to institutional practices, and the uptake of inclusive approaches. Follow-up studies over time would provide deeper insights into how the game experience translates into real-world professional contexts and whether its effects persist beyond the immediate workshop setting. Exploring the effectiveness of integrating explicit debriefing strategies focused on bias awareness within the game sessions also warrants further attention. Investigating the transferability of game-based interventions in different institutional contexts and with diverse staff populations will continue to contribute to the growing body of knowledge on ludic learning for professional development and the advancement of equity and inclusion in higher education. As part of this effort, we aim to make *Far From Home* available as an open educational resource to support wider adoption, adaptation, and future evaluation across diverse settings.

*Far From Home* offers a compelling model for leveraging game-based learning to foster empathy and reveal institutional biases. While the concept of *contrast commitment* emerged from a single participant’s account, it opens a theoretically promising direction for exploring how peer behaviours can reflexively reinforce equitable values—particularly within shared, emotionally engaging experiences. As a lens on interpersonal influence and bias awareness, this concept may extend beyond gameplay to other professional development settings, offering a foundation for future research into behavioural reinforcement within equity and inclusion initiatives. This study’s broader findings, alongside its acknowledged limitations, support the integration of playful, reflective approaches into university staff development. To support wider institutional adoption, we suggest embedding the game within existing EDI or induction pathways, offering facilitation guidance, and exploring hybrid delivery formats. In this way, *Far From Home* contributes not only a specific tool but a broader argument for ludic methods as catalysts for inclusive cultural change in higher education.

## Figures and Tables

**Figure 1 behavsci-15-00820-f001:**
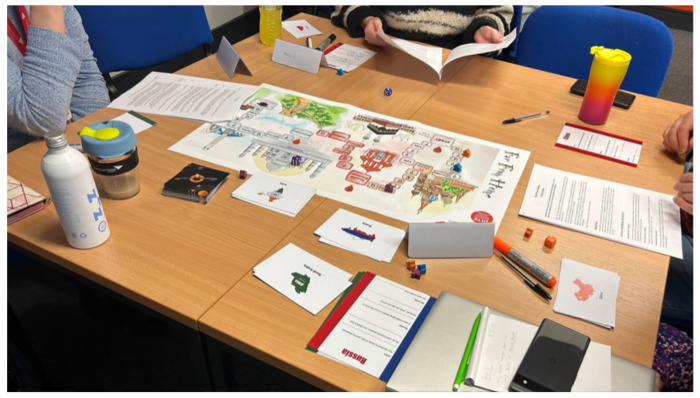
A picture taken during a game session where three players were present.

**Figure 2 behavsci-15-00820-f002:**
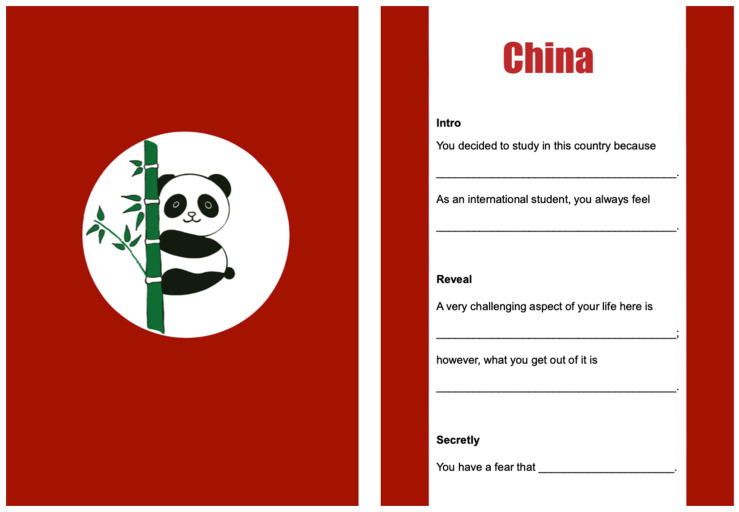
Nationality card of China with prompts to help players create their avatars.

**Figure 3 behavsci-15-00820-f003:**
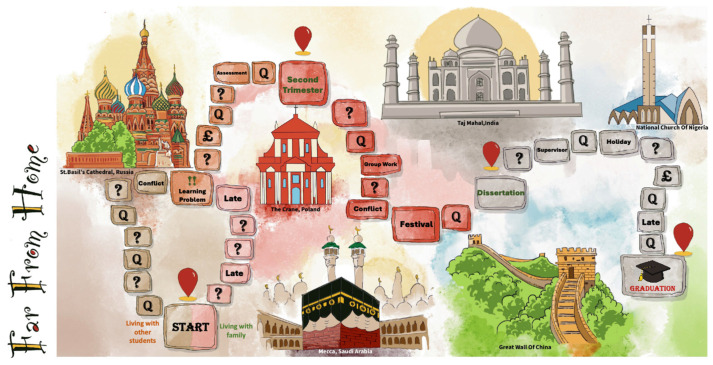
The game board resembling a typical master’s study experience.

**Figure 4 behavsci-15-00820-f004:**
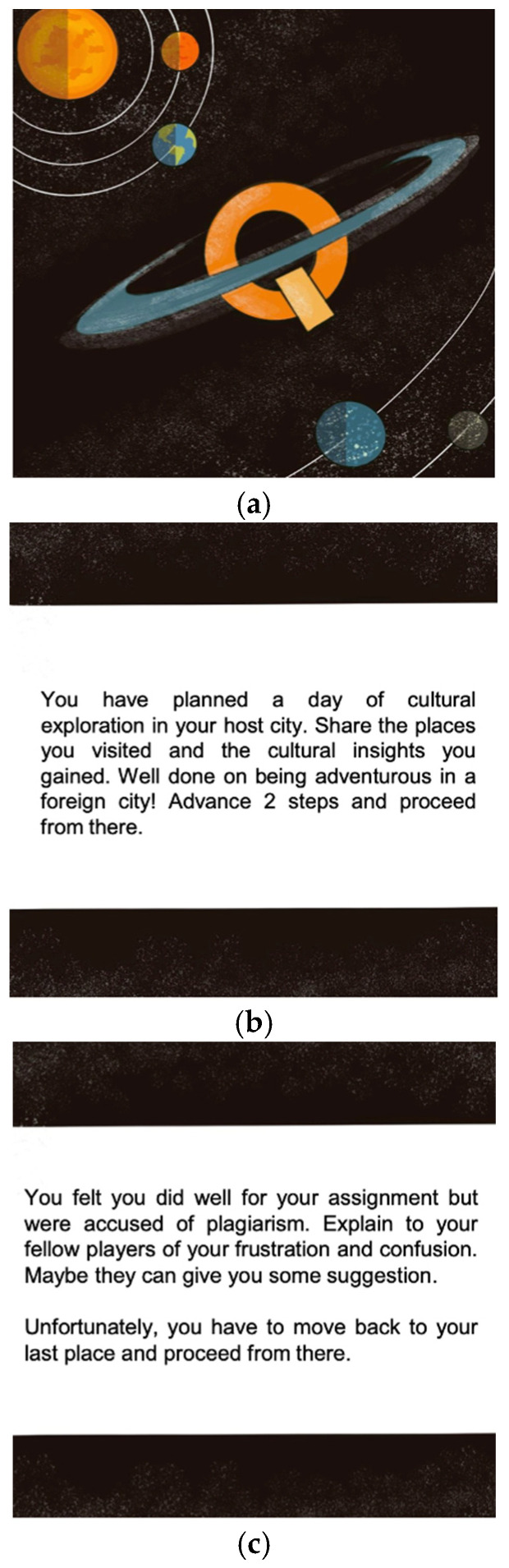
(**a**) The back of a quest card. (**b**) An example of a quest card. The quest asks the player to describe a cultural exploration day they have had, and the player obtains a reward for being adventurous in a foreign city. (**c**) An example of a quest card. The quest card describes a scenario where the player has been accused of plagiarism. The player is asked to share their confusion and frustration with the fellow players and obtain suggestions. The player is then punished by moving to the last place on the game board, thereby simulating the lack of power an international student may experience when facing an academic decision.

**Table 1 behavsci-15-00820-t001:** The mean scores and standard deviations for participant survey responses following the game session (scale: 1 = strongly disagree, 5 = strongly agree; overall experience: 1 = lowest, 5 = highest).

Survey Questions (From 1 Strongly Disagree to 5 Strongly Agree)	Mean	STDV
Playing the game reminds me of my time studying at school or university.	2.59	1.14
Playing the game makes me understand more of international students’ struggles studying and living in the UK.	4.41	0.68
I find the game instruction clear and easy to understand.	4.15	0.7
I find the game session well organised.	4.63	0.49
I find the game too hard.	1.81	0.66
I think the game is fun.	4.67	0.55
How would you rate your overall gaming experience from 1 the lowest to 5 the highest?	4.46	0.64

## Data Availability

The data that support the findings of this study are not publicly available due to privacy and ethical restrictions. Anonymised excerpts of qualitative data (e.g., de-identified quotations) can be made available upon reasonable request to the corresponding author.

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
