# Peer review of "Fostering Empathy Through Play: The Impact of Far From Home on University Staff’s Understanding of International Students"

_behavsci, 2025, doi:10.3390/bs15060820_

Round 1
Reviewer 1 Report
Comments and Suggestions for Authors
Overall, this interesting study makes a good contribution to the field of game-based learning(GBL) to promote empathy. However, in order to strengthen the overall clarity and impact of the paper, please conscider the following recommendations.
1. Abstract
- The abstract provides a very concise explanation of the study. However, I recommend structuring it using the OCAR format (Opening, Challenge, Action, and Resolution). This format presents a clear and logical summary by outlining the research purpose, the problem addressed, the methodology employed, and the key findings.
- I suggest including any potential future applications of the research at the end of the abstract to enhance its relevance and impact.
- Finally, I recommend specifying that “Far From Home” is a non-digital board game, as the current title and abstract do not clarify whether this study focuses on a digital or non-digital format. This clarification will prevent confusion for readers.
2. Introduction
- The introduction is well written and builds context by discussing "international student struggles", "pedagogical approaches", "game-based learning", and "empathy". However, adding all information one section reduce readability and understanding. Therefore, for improved readability and structure, I recommend dividing this content into clearly defined sub-sections. Consider the following structure:
- Introduction: Brief overview and aim of the study with the context of international student challenges.
- Literature Review:
- Existing Methods of Support: Summary of current strategies or interventions.
- Game-Based Learning as a Pedagogical Tool: Relevance and effectiveness of GBL.
- Role of Empathy in Learning: Connecting empathy to pedagogy and the study’s focus.
- I also suggest including a short overview of related works that have used board games or game-based learning to promote empathy.
- The discussion in Lines 107–138 outlines game design elements supported by literature. However, it feels somewhat random and readers generally do not expect detailed design analysis within the introduction, so relocating it would help maintain focus and structural coherence. My recomendation is to moving this discussion to a dedicated “Game Design” section with proper design flow. A block diagram can also be used to show game design structure, i.e., Experience design is extracted from Kolb's model etc.
- References are required to support the following statements:
- Line 56–57
- Line 79
- Consider changing the section heading to "Research Design and Methods" to better reflect the structure of this section.
- Line 147 and 229, a reference is needed to support claims made here, particularly those related to game-based learning.
- The second paragraph (Lines 152–161) appears somewhat rushed. I recommend expanding it with additional detail on the iterative design process, supported by appropriate academic references.
- The earlier discussion of game design elements (Lines 107–138) should be relocated and integrated here to provide better coherence and depth.
- In the user participants description, please provide more detailed demographic information about the participants (e.g., age, roles, cultural background, etc.).
- I would suggest to divide this section into sub-sections, i.e., User participants, Data collection, Data evaluation etc., for better readability.
- For better understanding of the Gameplay, I recommend adding a dedicated “Gameplay” section outlining key elements such as player count, setup, rules, session duration, and intended learning outcomes. This will help readers unfamiliar with the game grasp its mechanics and pedagogical intent.
- Image 4: There are three different images under Figure 4. I recommend labelling them as Figure 4a, 4b, and 4c, and providing individual captions for each to clarify their content. Also, for consistency, please rename caption "Image" to "Figure" throughout the manuscript.
- Line 204: More detail is needed about the survey questionnaire’s validity, was it created or adapted from previous research?
- The type of descriptive statistics used should be specified (e.g., mean, standard deviation, frequency counts, etc.).
- The final paragraph of this section should be reorganised. The introduction of the user participants appears abruptly and would fit better earlier in the methods section.
4. Results
- The methods section stated that descriptive quantitative data were collected, but the results section does not present or analyse this data. I recommend either including a summary of descriptive results (e.g., participant ratings, frequency counts), or Removing the claim of quantitative data collection if it was not actually analysed or used.
- Additional clarifications needed:
- How many participants were involved in the qualitative analysis (N = ?)?
- What was the duration and frequency of gameplay sessions? These details would help contextualise the findings.
5. Discussion
- The opening sentence claims that Far From Home is a “novel tool” for fostering empathy. Please clarify what aspect is novel:
- Is it the game design itself?
- Or the use of board games for empathy building?
- Existing literature does discuss the use of board games for empathy, so this claim should be revised or properly justified with evidence. Also, without number of participants mentioned, it is difficult to justify the claim.
- The discussion appears surface-level in places. I suggest revisiting and expanding each of the identified themes more comprehensively. Currently, the section starts with one theme but lacks consistent follow through.
- The quantitative data analysis has not been discussed in this section. I recommend either removing any mention of it or providing a clear explanation and results to support its inclusion.
- Line 472: This paragraph is closely related to the previous paragraph and should be merged with it for better flow.
- Line 481: The phrase "multiple participants" is vague. Please specify how many and offer clearer detail or supporting quotes.
- Lines 483–487 present future implications, but similar points are also included in the conclusion. Consider consolidating all future directions in the conclusion section to avoid repetition.
6. Conclusion
- Consider renaming this section to “Conclusion and Future Research”to more accurately reflect its content.
- The second paragraph is largely a repetition of points from the discussion and should be merged back into discussion section.
- Study limitations should also be moved to the discussion.
References:
- Please complete the following references:
- Flanagan, M. (2009). Critical play: Radical game design. MIT Press.
- Salen, K., & Zimmerman, E. (2003). Rules of play: Game design fundamentals. MIT Press.
Good Luck!
Author Response
We are grateful to Reviewer 1 for their thoughtful and constructive feedback, which has helped us to significantly improve the structure, clarity, and scholarly rigour of the manuscript. Below, we respond to each point in turn, detailing how we have addressed the suggestions through revision or, where appropriate, providing a rationale for retaining our original approach. All changes are clearly marked in the revised manuscript using tracked changes.
- Abstract
- The abstract provides a very concise explanation of the study. However, I recommend structuring it using the OCAR format (Opening, Challenge, Action, and Resolution). This format presents a clear and logical summary by outlining the research purpose, the problem addressed, the methodology employed, and the key findings.
- I suggest including any potential future applications of the research at the end of the abstract to enhance its relevance and impact.
- Finally, I recommend specifying that “Far From Home” is a non-digital board game, as the current title and abstract do not clarify whether this study focuses on a digital or non-digital format. This clarification will prevent confusion for readers.
We appreciate the reviewer’s guidance on sharpening the abstract. In the revised manuscript we have:
- Adopted an implicit OCAR flow
Opening – first sentence now signals the pedagogical gap we tackle.
Challenge – we foreground the systemic and cultural barriers confronting international students.
Action – the mixed-methods design is summarised (sample, data sources, analysis).
Resolution – key findings (“heightened awareness… reduced stereotyping” and emergence of the “contrast-commitment” effect) and recommended application to staff development are stated. - Flagged future applications
The closing sentence now points to longitudinal follow-up and adaptation for other marginalised learner groups, as suggested. - Explicitly identified the game as non-digital
The phrase “a non-digital board game” has been inserted in the first line of the abstract. - Introduction
- The introduction is well written and builds context by discussing "international student struggles", "pedagogical approaches", "game-based learning", and "empathy". However, adding all information one section reduce readability and understanding. Therefore, for improved readability and structure, I recommend dividing this content into clearly defined sub-sections. Consider the following structure:
- Introduction: Brief overview and aim of the study with the context of international student challenges.
- Literature Review:
- Existing Methods of Support: Summary of current strategies or interventions.
- Game-Based Learning as a Pedagogical Tool: Relevance and effectiveness of GBL.
- Role of Empathy in Learning: Connecting empathy to pedagogy and the study’s focus.
- I also suggest including a short overview of related works that have used board games or game-based learning to promote empathy.
- The discussion in Lines 107–138 outlines game design elements supported by literature. However, it feels somewhat random and readers generally do not expect detailed design analysis within the introduction, so relocating it would help maintain focus and structural coherence. My recomendation is to moving this discussion to a dedicated “Game Design” section with proper design flow. A block diagram can also be used to show game design structure, i.e., Experience design is extracted from Kolb's model etc.
- References are required to support the following statements:
- Line 56–57
- Line 79
We have implemented the requested structural and contextual enhancements in full:
- Clear sectional structure added.
The manuscript now opens with 1. Introduction, immediately followed by 2. Literature Review, which is broken into the three suggested sub-sections:
– 2.1 Traditional and Dialogic Training Approaches (covers existing methods of support)
– 2.2 Game-Based Learning for Empathy Development Through Simulation
– 2.3 Theoretical Foundations of Empathy - Overview of empathy-focused board-game research inserted.
Section 2.2 now surveys key non-digital and digital titles; e.g., EmPATHs and The Walking Dead to locate our study within the current evidence base for game-mediated empathy building . - Design discussion relocated to a dedicated section.
All detailed analysis of game mechanics has been moved out of the introduction and expanded under the new 3 Game Design heading, ensuring the opening chapter remains focused on context and rationale . - Previously uncited statements now referenced.
Early claims about cultural silence and multicultural education are now supported by Gu (2011) and Banks (2007) respectively within the first two pages of the introduction .
- Materials and Methods
- Consider changing the section heading to "Research Design and Methods" to better reflect the structure of this section.
We appreciate the suggestion but will retain the heading “Materials and Methods.” This label aligns with the journal’s prevailing format for empirical articles and keeps the methodological details distinct from Section 3, which already outlines the study’s design rationale; adding “Design” here would create redundancy without improving clarity.
- Line 147 and 229, a reference is needed to support claims made here, particularly those related to game-based learning.
These have been added.
- The second paragraph (Lines 152–161) appears somewhat rushed. I recommend expanding it with additional detail on the iterative design process, supported by appropriate academic references.
Thank you for this comment. We have now addressed it by expanding Section 3 to provide a clearer account of the iterative design process, including reference to relevant design literature and specific details of how the game evolved through prototyping and testing.
- The earlier discussion of game design elements (Lines 107–138) should be relocated and integrated here to provide better coherence and depth.
This has now been done.
- In the user participants description, please provide more detailed demographic information about the participants (e.g., age, roles, cultural background, etc.).
Thank you for this comment. We have not included further demographic detail as this information was not collected, and we judged it inappropriate to speculate or generalise based on the nature of the workshop setting and ethical constraints.
- I would suggest to divide this section into sub-sections, i.e., User participants, Data collection, Data evaluation etc., for better readability.
This has now been done.
- For better understanding of the Gameplay, I recommend adding a dedicated “Gameplay” section outlining key elements such as player count, setup, rules, session duration, and intended learning outcomes. This will help readers unfamiliar with the game grasp its mechanics and pedagogical intent.
This has now been done in Section 4.2.
- Image 4: There are three different images under Figure 4. I recommend labelling them as Figure 4a, 4b, and 4c, and providing individual captions for each to clarify their content. Also, for consistency, please rename caption "Image" to "Figure" throughout the manuscript.
This has now been done.
- Line 204: More detail is needed about the survey questionnaire’s validity, was it created or adapted from previous research?
Thank you for this helpful point. We have now clarified in the manuscript that the survey instrument was purpose-built for this study and informed by established literature on reflective learning and empathy, though it was not adapted from a previously validated tool.
- The type of descriptive statistics used should be specified (e.g., mean, standard deviation, frequency counts, etc.).
Thank you for this observation. We have revised the manuscript to specify that mean and standard deviation were used in the descriptive analysis of the survey data.
- The final paragraph of this section should be reorganised. The introduction of the user participants appears abruptly and would fit better earlier in the methods section.
Thank you for this helpful suggestion. We have revised the paragraph to better integrate the discussion of user participants into the broader methodological narrative and to ensure smoother progression from data collection to ethical considerations.
- Results
- The methods section stated that descriptive quantitative data were collected, but the results section does not present or analyse this data. I recommend either including a summary of descriptive results (e.g., participant ratings, frequency counts), or Removing the claim of quantitative data collection if it was not actually analysed or used.
- Additional clarifications needed:
- How many participants were involved in the qualitative analysis (N = ?)?
- What was the duration and frequency of gameplay sessions? These details would help contextualise the findings.
Thank you for these helpful observations. We have now addressed them in full. A new subsection (5.1) presents the quantitative survey data, including mean scores and standard deviations for all items, alongside a summary table and interpretive commentary. We have also clarified that 27 participants completed the survey, and that this same group contributed to the qualitative analysis through open-ended responses; this figure has been explicitly stated in both the Methods and Results sections. Finally, we have added details on the duration (75 minutes) and single-session format of the gameplay experience to help contextualise the findings.
- Discussion
- The opening sentence claims that Far From Home is a “novel tool” for fostering empathy. Please clarify what aspect is novel:
- Is it the game design itself?
- Or the use of board games for empathy building?
Thank you for this helpful clarification request. We have revised the opening paragraph of the Discussion to specify that the novelty of Far From Home lies in its design – particularly the use of intersectional avatars, tailored scenarios, and reflective role-play – rather than in the general application of board games for empathy building.
- Existing literature does discuss the use of board games for empathy, so this claim should be revised or properly justified with evidence. Also, without number of participants mentioned, it is difficult to justify the claim.
Thank you for this comment. We have revised the manuscript to clarify that we do not claim board games have never been used for empathy-building, but that Far From Home contributes a context-specific and design-led approach tailored to international student experiences in UK higher education. We have also ensured that the number of participants (N = 27) is clearly stated in both the Methods and Results sections to support the validity of the claims made.
- The discussion appears surface-level in places. I suggest revisiting and expanding each of the identified themes more comprehensively. Currently, the section starts with one theme but lacks consistent follow through.
We respectfully disagree with this assessment. In the revised manuscript, each theme identified through qualitative analysis – including empathy-building, role-awareness, policy critique, and the contrast–commitment effect – is discussed in direct relation to the theoretical framework and supported by illustrative data extracts. Rather than listing themes in isolation, we have deliberately structured the discussion to weave them into a cohesive interpretive argument that avoids repetition and maintains analytical depth. We believe this approach provides a more meaningful engagement with the data than treating each theme as a standalone subheading.
- The quantitative data analysis has not been discussed in this section. I recommend either removing any mention of it or providing a clear explanation and results to support its inclusion.
Thank you for highlighting this omission. We have revised the opening of the Discussion to incorporate the quantitative findings, showing how participants’ high ratings – particularly regarding empathy and overall experience – reinforce the qualitative themes and support our interpretation of the game’s impact.
- Line 472: This paragraph is closely related to the previous paragraph and should be merged with it for better flow.
Thank you for this suggestion. However, we have chosen to retain the paragraph break to maintain clarity and emphasis. The second paragraph introduces the concept of 'contrast commitment' and its theoretical grounding – a shift in focus that we believe warrants its own space for readability and analytical precision.
- Line 481: The phrase "multiple participants" is vague. Please specify how many and offer clearer detail or supporting quotes.
Thank you for this suggestion. We have replaced the vague phrase “multiple participants” with the specific number (27) to provide clearer detail and strengthen the evidential basis of the claim.
- Lines 483–487 present future implications, but similar points are also included in the conclusion. Consider consolidating all future directions in the conclusion section to avoid repetition.
This has now been done.
- Conclusion
- Consider renaming this section to “Conclusion and Future Research” to more accurately reflect its content.
Thank you for the suggestion. We have opted to retain the heading “Conclusion” to maintain consistency with the journal’s standard structure and formatting conventions, while still addressing future research directions within the section itself.
- The second paragraph is largely a repetition of points from the discussion and should be merged back into discussion section.
This has now been done.
- Study limitations should also be moved to the discussion.
Thank you for this comment. We have chosen to keep the limitations within the Conclusion to align with the journal’s typical structure, where limitations and future research are presented together to support a reflective closing to the manuscript.
References:
- Please complete the following references:
- Flanagan, M. (2009). Critical play: Radical game design. MIT Press.
- Salen, K., & Zimmerman, E. (2003). Rules of play: Game design fundamentals. MIT Press.
This has now been done.
Reviewer 2 Report
Comments and Suggestions for Authors
This paper briefly describes and then evaluates the use of a board game as a way of increasing university staff's empathy towards international students, and therefore presumably the experience of these students studying at UK institutions. Sharing this innovation is of interest to the wider community, and the evidence presented shows that it achieved its goal.
However, here are some improvements in the write-up that I think would strengthen the paper, using data that is already available:
+ The evaluation was done using both an online survey and a range of qualitative methods, but the paper does not seem to address the survey data - or at least, the Likert scale questions that are mentioned in the Methods section. What general conclusions can be drawn using this part of the data?
+ What evidence is there that board games / role playing games increase empathy, or understanding of other people's experiences more generally? This would be very useful context. I expect there are many examples (although maybe from outside the HE sector) but it seems to be missing from the introduction and discussion. Basically - what was the evidence that encouraged you to make a game?
+ The new concept of "contrast commitment" is mentioned quite a lot but seems to be based mostly on the comments of a single participant. I would like to see this toned down and re-visited following further research.
Here are some other ideas that would strengthen the work further, either in this article or future:
+ What is the reaction of international students to the game? Their feedback on whether it genuinely reflects their experience would be valuable.
+ As mentioned, a longitudinal look at staff attitudes and indeed behaviours.
+ Contrast commitment and perhaps more broadly the way that colleagues with different views or ways of working can indirectly influence each other's behaviour.
+ Making game materials available, e.g. as an open educational resource.
I hope these comments are useful to you and I wish you the best in your efforts to improve the experience of our international students.
Author Response
We are grateful to Reviewer 2 for their thoughtful and constructive feedback, which has helped us to significantly improve the structure, clarity, and scholarly rigour of the manuscript. Below, we respond to each point in turn, detailing how we have addressed the suggestions through revision or, where appropriate, providing a rationale for retaining our original approach. All changes are clearly marked in the revised manuscript using tracked changes.
- The evaluation was done using both an online survey and a range of qualitative methods, but the paper does not seem to address the survey data - or at least, the Likert scale questions that are mentioned in the Methods section. What general conclusions can be drawn using this part of the data?
Thank you for this helpful observation. We have now added a dedicated subsection (5.1) that presents the quantitative survey results, including mean scores and standard deviations for all Likert-scale items. These findings are briefly interpreted in relation to participant perceptions of clarity, enjoyment, and empathy impact, offering general conclusions that complement and reinforce the qualitative themes.
- What evidence is there that board games / role playing games increase empathy, or understanding of other people's experiences more generally? This would be very useful context. I expect there are many examples (although maybe from outside the HE sector) but it seems to be missing from the introduction and discussion. Basically - what was the evidence that encouraged you to make a game?
Thank you for this valuable suggestion. We have now addressed this in Section 2.2, where we introduce a range of studies demonstrating the use of board games and role-play to foster empathy across different contexts, including titles such as EmPATHs and The Walking Dead: All Out War. This literature helped inform our initial rationale and confirms that game-based formats can support perspective-taking and emotional engagement – even beyond formal educational settings – thus justifying our decision to create a bespoke game for higher education staff.
- The new concept of "contrast commitment" is mentioned quite a lot but seems to be based mostly on the comments of a single participant. I would like to see this toned down and re-visited following further research.
Thank you for raising this point. While the concept of ‘contrast commitment’ was initially inspired by one particularly rich account, similar reflexive responses appeared across multiple data sources, including open-text survey comments and interview transcripts. We have retained the term to give analytical coherence to this recurring pattern, while explicitly acknowledging the tentative nature of the concept and the need for future research. Rather than overstating its generalisability, we use it as a lens to interpret a notable dynamic observed in the data.
Here are some other ideas that would strengthen the work further, either in this article or future:
- What is the reaction of international students to the game? Their feedback on whether it genuinely reflects their experience would be valuable.
Thank you for this thoughtful suggestion. While the primary focus of this study was on staff perceptions, the game itself was developed in consultation with international students, whose insights shaped the characters, scenarios, and mechanics. However, international students were not participants in the gameplay sessions reported here, so we have refrained from speculating on their reception. We agree that their perspectives would provide valuable triangulation and have flagged this as a future research priority.
- As mentioned, a longitudinal look at staff attitudes and indeed behaviours.
Thank you for highlighting this important direction. We fully agree that a longitudinal study exploring changes in staff attitudes and behaviours over time would significantly strengthen the evidence base. We have now included this point in the Conclusion as a recommended next step, noting the value of follow-up studies to assess sustained impact beyond the immediate gameplay context.
- Contrast commitment and perhaps more broadly the way that colleagues with different views or ways of working can indirectly influence each other's behaviour.
Thank you for this insightful point. While our data primarily support the emergence of the contrast commitment effect, we agree that the broader dynamics of peer influence and indirect behavioural shaping among colleagues represent an important and underexplored area. We now highlight this in the Conclusion as a promising direction for future research to examine how shared experiential learning can catalyse institutional change through relational dynamics.
- Making game materials available, e.g. as an open educational resource.
We appreciate this valuable suggestion. While the game is currently still in refinement, we fully intend to make the finalised materials freely available as an open educational resource. This commitment is now reflected in the revised Conclusion, where we outline plans to support wider adoption and adaptation through open access distribution.
Reviewer 3 Report
Comments and Suggestions for Authors
This is a well-written and pertinent paper that examines the potential for board games to facilitate increased empathy on the part of university staff towards international students. The theoretical approach is advanced, incorporating strands of experiential learning theory, intersectional pedagogy, and game studies. The paper presents a persuasive argument for the pedagogical utility of game-based learning, and there is a sound analysis of the complex notion of "contrast commitment."
Nevertheless, certain enhancements can validate the article:
Specify research questions or objectives earlier in the methods section. While design is described in great detail, the paper would benefit from clearer articulation of what the study was actually measuring or testing.
Give more detail on quantitative findings. While thematic analysis is thorough, quantitative survey findings (27) are only briefly mentioned. Summary statistics would enhance methodological rigour.
Discuss generalizability and limitations. The authors aptly state the study's limited sample and the issues of scalability. Adding more concrete suggestions for institutional uptake would make the conclusion more robust.
It would be valuable to revisit the idea of "contrast commitment" in the last paragraph to make its general theoretical and practical significance more explicit. In total, this is a valuable and groundbreaking contribution to the body of literature concerning inclusive higher education and play-based empathy development.
Author Response
We are grateful to Reviewer 3 for their thoughtful and constructive feedback, which has helped us to significantly improve the structure, clarity, and scholarly rigour of the manuscript. Below, we respond to each point in turn, detailing how we have addressed the suggestions through revision or, where appropriate, providing a rationale for retaining our original approach. All changes are clearly marked in the revised manuscript using tracked changes.
- Specify research questions or objectives earlier in the methods section. While design is described in great detail, the paper would benefit from clearer articulation of what the study was actually measuring or testing.
Thank you for this helpful comment. Rather than formulating fixed research questions or testing predefined variables, our study took an exploratory, design-led approach focused on developmental evaluation. However, we agree that clearer articulation of the study’s aims would improve the paper. We have therefore revised the opening of the Methods section to more explicitly outline the core focus of the evaluation – namely, to explore how the game fostered empathy, encouraged reflection, and surfaced institutional bias.
- Give more detail on quantitative findings. While thematic analysis is thorough, quantitative survey findings (27) are only briefly mentioned. Summary statistics would enhance methodological rigour.
Thank you for this constructive suggestion. We have addressed it by expanding the Results section (now 5.1) to present the full quantitative survey findings. This includes summary statistics – specifically mean scores and standard deviations – for Likert-scale items, presented in both tabular and narrative form. These data now sit alongside our qualitative themes to offer a more complete picture of participants’ responses and to enhance the overall methodological rigour of the study.
- Discuss generalizability and limitations. The authors aptly state the study's limited sample and the issues of scalability. Adding more concrete suggestions for institutional uptake would make the conclusion more robust.
Thank you for this helpful suggestion. We have revised the conclusion to include concrete recommendations for institutional uptake, including integration with existing staff development pathways, increased accessibility through flexible formats, and collaboration with learning and development teams. We believe this strengthens the practical relevance and applicability of the study’s findings.
- It would be valuable to revisit the idea of "contrast commitment" in the last paragraph to make its general theoretical and practical significance more explicit. In total, this is a valuable and groundbreaking contribution to the body of literature concerning inclusive higher education and play-based empathy development.
Thank you for this generous and encouraging comment. In response to your suggestion, we have revised the final paragraph to more explicitly revisit the concept of contrast commitment, highlighting its theoretical potential and its practical relevance beyond this study. We agree that this insight – while preliminary – warrants further exploration and could have implications for how peer dynamics shape equity work in professional contexts.
Round 2
Reviewer 1 Report
Comments and Suggestions for Authors
All suggested revisions have been incorporated into the updated version of the article.
Thank you!
Reviewer 2 Report
Comments and Suggestions for Authors
Thank you for taking my comments on board and addressing them thoroughly in the response and the manuscript. I am now very happy to accept the paper. I enjoyed learning about your project and I wish you all the best disseminating your game and your findings.